# A Gap of Patients with Infective Endocarditis between Clinical Trials and the Real World

**DOI:** 10.3390/jcm12041566

**Published:** 2023-02-16

**Authors:** Nobuhiro Asai, Yuichi Shibata, Jun Hirai, Wataru Ohashi, Daisuke Sakanashi, Hideo Kato, Mao Hagihara, Hiroyuki Suematsu, Hiroshige Mikamo

**Affiliations:** 1Department of Clinical Infectious Diseases, Aichi Medical University Hospital, Nagakute 480-1195, Aichi, Japan; 2Department of Infection Control and Prevention, Aichi Medical University Hospital, Nagakute 480-1195, Aichi, Japan; 3Department of Pathology, University of Michigan, Ann Arbor, MI 48105, USA; 4Division of Biostatistics, Clinical Research Center, Aichi Medical University Hospital, Nagakute 480-1195, Aichi, Japan; 5Department of Pharmacy, Mie University Hospital, Tsu 514-8507, Mie, Japan; 6Department of Clinical Pharmaceutics, Division of Clinical Medical Science, Mie University Graduate School of Medicine, Tsu 514-8507, Mie, Japan; 7Department of Molecular Epidemiology and Biomedical Sciences, Aichi Medical University, Nagakute 480-1195, Aichi, Japan

**Keywords:** infective endocarditis, randomized clinical trial, real-world, evidence-based medicine

## Abstract

**Introduction:** A randomized control trial (RCT) is considered to be the highest level in the Evidence-Based Medicine (EBM) pyramid. While EBM is essential to make a practical tool such as a prognostic guideline, it has been unclear how many patients in the real world can be eligible for a randomized control trial (RCT). **Patients and method:** This study was performed to clarify if there is a difference in patients’ profiles and clinical outcomes between the patients eligible and not eligible for any RCT. We reviewed all IE patients at our institute between 2007 and 2019. The patients were divided into two groups: those eligible for RCTs (RCT appropriate group) and those who were not (RCT inappropriate group). Exclusion criteria for clinical trials were set based on previous clinical trials. **Results:** A total of 66 patients were enrolled in the study. The median age was 70 years (range 18 to 87 years), and 46 (70%) were male. Seventeen (26%) of the patients were eligible for RCTs. Comparing the two groups, patients in the RCT appropriate group were younger and had fewer comorbidities. The disease severity was milder in the RCT appropriate groups than in the RCT inappropriate groups. Patients in the RCT appropriate group showed significantly longer overall survival times than those in the RCT inappropriate group (Log-Rank test, *p* < 0.001). **Conclusions:** We found a significant gap in patients’ characteristics and clinical outcomes between the groups. Physicians should be aware that RCT can never reflect the real-world population.

## 1. Introduction

“Evidenced based medicine (EBM) is established primarily based on the results of clinical trials. Because of this, clinical trials are considered to be one of the most important undertakings and put at the top of priority among physicians in constructing therapeutic strategies [1,2]”. While a randomized control trial (RCT) is the highest evidence level, there must be a gap of patients between clinical trials such as RCTs and patients in the real world, as we previously mentioned [2,3]. Since there was a large gap of patients’ characteristics in between the real world and clinical trials, candidemia and community-onset pneumonia patients in the real world showed a poorer outcome than those who can be enrolled in clinical trials [2,3]. Despite this, some clinicians tend to be overconfident in the results of RCTs and unfortunately, they are not aware of or overlook the fact that there is a difference in patient’s profiles between the real world and a clinical trial.

Infective endocarditis (IE) is one of the most serious septic infectious diseases showing a high in-hospital mortality of 16–26% [4,5,6,7], although antibiotic therapy and surgical treatment were advanced. Due to an advancing aging society, more elderly patients with IE are estimated, and they may not tolerate the standard treatment such as surgery due to a poor condition. We conducted this study to examine the difference of patient’s profiles and outcome between the real world and a clinical trial. This is the first report documenting that an IE patient’s profile and outcomes in the real world are quite different from those in a clinical trial.

## 2. Patients and Methods

### 2.1. Participants

Our institute, which is a 900-bed tertiary care center, is located in the countryside at Aichi Prefecture in the central area of Japan. For the purpose of estimating how many infective endocarditis (IE) patients in our institute could participate in any randomized clinical trials for the treatment of IE, we reviewed all IE patients who were admitted in our hospital between 2007 and 2019. Patients diagnosed as definite IE according to the modified Duke criteria or by surgical procedure were included in this study. Patients diagnosed as having possible IE according to the modified Duke criteria [8] were excluded in this study.

The severity of IE was determined by a systemic inflammatory response syndrome (SIRS) score [9], quick SOFA (qSOFA), and SOFA scores [10]. The Charlson comorbidity index (CCI) evaluated patents’ comorbidity [11]. We previously reported that SOFA score was a good predictive marker among IE patients for in-hospital death [7]. Thus, we evaluated severity of IE by SOFA score.

The patients were divided into two groups: patients who were eligible for clinical trials (RCT appropriate group), and those who were not (RCT inappropriate group). Then, patients’ characteristics (age and sex); pathogens isolated; clinical outcomes such as the treatments and/or results; 30-day or in-hospital mortality; and the reasons of exclusion from the clinical trial were evaluated.

### 2.2. Definitions

Vascular phenomena included major arterial emboli, septic pulmonary infarcts, mycotic aneurysms, intracranial hemorrhages, conjunctival hemorrhages, and Janeway lesions. Immunological phenomena included glomerulonephritis, Osler’s nodes, Roth spots, decrease in complements in serum level, and rheumatoid factor positivity. Vascular and immunological phenomena were regarded as absent when none of these phenomena were documented. Antibiotic treatment was classified as appropriate or as inappropriate when the identified pathogens were sensitive and resistant, respectively, to the initially prescribed antibiotics. The antibiotic susceptibility was assessed with minimum inhibitory concentration testing according to the guidelines of the Clinical and Laboratory Standards Institute [12]. Persistent bacteremia was defined as any blood culture growing the same microorganism as the index culture ≥48 h after the start of the antimicrobial therapy [13]. Disseminated intravascular [13] coagulation (DIC) was diagnosed according to the disseminated intravascular coagulation diagnostic criteria established by the Japanese Association for Acute Medicine (JAAM DIC diagnostic criteria) [14]. These definitions were consistent with the previous study’s criteria [2].

### 2.3. Exclusion Criteria for Randomized Control Trial

Exclusion criteria commonly used in the past ordinary clinical trials are as follows [15,16,17]:Age < 18 years, >80 years.Coexisting comorbidities or medical conditions that could make the evaluation of infective endocarditis difficult such as severe liver dysfunction, severe renal dysfunction, or HIV/AIDS. (Severe liver dysfunction was defined as serum total bilirubin, or aspartate aminotransferase/alanine aminotransferase > the upper limit of the normal reference range ×3. Severe renal dysfunction was defined as Creatinine clearance < 30 mL/min.) Unassessable malignancies were defined as any terminal stage malignancy) or any with metastatic lesions. Unassessable diabetes mellitus was defined as serum-hemoglobin A1c (NGSP) ≧ 7.0%.Having prosthetic valve involvement.Receiving immunosuppressive therapy due to any cause.Receiving chemotherapy for malignancy.Receiving palliative therapy for malignancy.Receiving hemodialysis due to any cause.Having other complicated infection such as mycobacterial tuberculosis.Inability to give full informed consent such as cognitive impairment due to any cause.Causative pathogens are fungi.Multiple pathogens identified.Need of prolonged antibiotic therapy due to spondylodiscitis or other septic complication.Patients who need urgent cardiac surgery but are considered inoperable due to high surgical risks.Poor prognosis (anticipated life expectancy <90 days or patients who are not expected to survive until the end of the trial).Pregnant or breastfeeding women.

### 2.4. Statistical Analysis

The data for categorical variables are expressed as percentages and continuous variables as mean ± standard deviation (SD). Chi-square or Fisher’s exact test (two-tailed) was used to compare categorical variables and unpaired Student’s *t*-test or Mann–Whitney U test to compare continuous variables. Statistical analyses involved the use of SPSS version 26 for Windows (SPSS Inc., Chicago, IL, USA). Kaplan–Meier analyses were carried out using Graph Pad Prism v 9.3.1. Overall survival time (OS) was calculated from the date of diagnosis until the date of death from any cause. Generalized Wilcoxon-test and Log-Rank-test evaluated significance. A *p*-value < 0.05 was considered statistically significant. This study was approved by the Institutional Review Board of Aichi Medical University Hospital.

## 3. Results

A total of 66 patients were enrolled in the study. Table 1 shows the patients’ characteristics and clinical outcomes. They were 46 males (70%), and the median age was 70 years (range 19 to 87 years). As for the infection site, mitral valve was most commonly seen in 35 (53%), followed by aortic valve in 20 (30%). Native valve and prosthetic valve involvement were found in 53 (80%) and 13 (20%), respectively. In terms of underlying diseases, heart disease was most frequently seen in 33 (50%), followed by diabetes mellitus in 21 (32%). The mean CCI was 2.1 (±2.2). As for clinical symptoms, immunological and vascular phenomena were seen in 23 (35%) and 47 (71%), respectively. In terms of initial antibiotic therapy, monotherapy and combination therapy were seen in 25 (38%) and 41 (62%), respectively. The most common monotherapy was penicillin in 10 (15%), followed by cephalosporin in 7 (11%). Combination with anti-MRSA agents was used most frequently in 27 (41%). The mean durations of hospital stay and antibiotic treatment were 64.2 days and 123.4 days, respectively. The 30-day and in-hospital deaths were seen in 8 (12%) and 16 (24%), respectively. Regarding pathogens, *Staphylococcus aureus* was identified most commonly in 23 (35%), followed by coagulase- negative streptococci in 12 (18%).

Seventeen of the patients (26%) were categorized into the RCT appropriate group, while 49 patients (74%) were in the RCT inappropriate group. When comparing the RCT appropriate and inappropriate groups, patients in the RCT appropriate group were younger, and had better PSs than the RCT inappropriate group. Less patients in the RCT appropriate group had malignancy (6% vs. 31%, *p* = 0.04) and showed lower mean CCIs than the RCT inappropriate group (0.9 vs. 2.5, *p* = 0.009). As for treatment, more surgery was performed in the RCT appropriate group than in the RCT inappropriate group. There were no differences in the regimens of initial antibiotic therapy between the groups.

Kaplan–Meier analyses showed that RCT appropriate group had a significantly longer OS than RCT inappropriate group (Figure 1A). In terms of performance status, the patients with PS 0-1 displayed a significant longer OS than those with PS 2-4 (Figure 1B). The RCT appropriate group with PS 0-1 had a longer OS than RCT inappropriate group with PS 0-1, even though there was no significance (*p* = 0.143 by *Log-Rank* test) (Figure 1C). Nevertheless, the RCT appropriate group with PS 2-4 showed an almost significantly longer OS than RCT inappropriate group (*p* = 0.07 by *Log-Rank* test) (Figure 1D).

Table 2 shows the reasons for not being appropriate for RCTs. The most common reason was to have received any palliative therapy in 17 (26%), followed by unassessed comorbidities in 13 (20%), and prosthetic valve involvements in 13 (20%).

## 4. Discussion

Since EBM is considered to be the highest level of medical practice, physicians tend to prioritize it in general practice, and some are overconfident in RCTs. As the data showed, EBM based on RCT is very limited in scope, and EBM has never reflected the real world as previous studies have already reported [2,18].” IE patients in the real world are quite different from those who can be enrolled in a clinical trial, showing an unfavorable outcome. Only 26% (17 of the 66) of patients were appropriate for RCT enrollment. Moreover, patients in the RCT inappropriate group showed significantly lower OSs than those in the RCT appropriate group. This remarkable trend was reproducible in previous studies [2,18]. In terms of mortality among IE patients, the mortality rate in RCT was much lower than our cohort, showing 2.9–18% of in-hospital mortality rate [17,19,20]. Differences in patients’ general conditions and strict entry criteria for the RCTs might have contributed to better outcomes than the real-world data. Tolerability to surgical treatment depends on whether patients have complications and poor general conditions. In our cohort, patients who received surgical treatment had more favorable outcomes than those who did not (in-hospital mortality 5% vs. 32%, *p* = 0.027). Patients who underwent surgical treatment had lower mean CCI scores than those who did not (mean CCI score ± standard deviation (SD) 0.7 (±1.2) vs. 2.6 (±2.3), *p* = 0.001), although there was no significant difference in PS between the groups (mean PS (±SD) 1.6 (±1.6) vs. 2.1 (±1.7), *p* = 0.308). These results are consistent with the fact that there is a large gap between eligible patients and those who are not eligible for RCT.

Guidelines for several diseases are helpful and useful for clinicians, particularly for those who do not specialize in the field, to help them make a rational decision in treating patients. We are concerned that clinicians may apply a biased EBM to patients who cannot be enrolled in a clinical trial. More than 50% of IE patients are generally indicated for surgical intervention and despite evident surgical indication that were present, surgery was not performed in 26–42% of patients [17,21]. Those who were not indicated for surgical treatment showed a high mortality rate of about 63% [21,22,23]. “Systemic embolism, which occurs in approximately one-third of patients with infective endocarditis and involves the central nervous system in up to 65%, is the second most common cause of death, following congestive heart failure, in this patient population [21]”. Although the indication between surgery and antibiotic therapy had no clear-cut distinction, embolism to the brain can become one reason that IE patients with involvement in the central nervous system are denied surgery. The frequency of detecting embolic lesions in the brain or spine may depend on the institute and country. The previous cohort from Japan showed that 93% of IE patients with neurological abnormality had intracranial abnormalities on head MRI (hemorrhage in three and abscess formation in three) [24]. The authors concluded that MRI should be promptly performed on IE patients. In our cohort, 61 of the 66 patients received head MRA, although 54 of the 66 patients (82%) had no central nervous symptoms. Antibiotic agents that have good penetration into the cerebrospinal fluid and brain tissue should be selected for preventing neurological complications in IE patients. Surprisingly, the sub-analysis showed no difference in the proportion of IE cases with or without brain embolism who underwent surgery (35% vs. 24%, *p* = 0.561). As for the prognosis of IE patients, there was no significant difference in in-hospital mortality rate between with and without the brain embolism.

In Japan, there is a universal nationwide healthcare insurance system, which allows everyone to easily have access to any medical institute. All patients will be able to receive head or spine MRIs at a low cost. This specific environment might improve the prognosis of the IE patients with brain embolisms compared to the previous cohort.

There are several limitations in the study. First, this is a retrospective study with a small sample size at a single-center institute. Second, the study included only patients with definite IE according to the modified Duke criteria or by surgical procedure. Some IE patients might not have been included in our study. Third, there might have been a career-bias in physicians and surgeons. This might have affected the results.

We conclude that there is a large gap in patients’ characteristics and clinical outcomes between IE patients who are eligible and not eligible for RCTs. Every physician should be aware that RCTs reflect only a small proportion of the real-world population, although RCTs are essential to advance medical treatments.

## Figures and Tables

**Figure 1 jcm-12-01566-f001:**
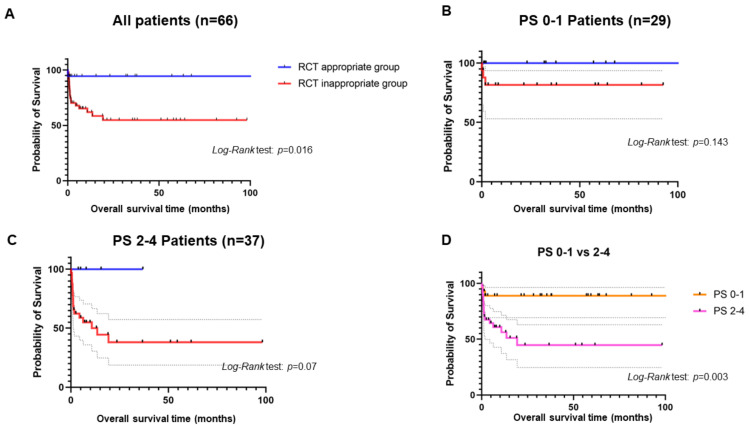
Kaplan–Meier curves. (**A**) shows the comparison of OSs between RCT appropriate and inappropriate groups among IE patients. (**B**) shows the comparison of OSs between RCT appropriate and inappropriate groups among IE patients with PS 0-1. (**C**) shows the comparison of OSs between RCT appropriate and inappropriate groups among IE patients with PS 2-4. (**D**) shows the comparison of OSs between IE patients with PS 0-1 and those with PS 2-4.

**Table 1 jcm-12-01566-t001:** Comparison of patients’ characteristics and outcomes among the RCT appropriate group and inappropriate group.

Variables	All Patients (*n* = 66)	RCT Appropriate Group (*n* = 17)	RCT Inappropriate Group (*n* = 49)	*p*-Value
Mean age (years ± SD)	64.7 ± 17.3	53.5 ± 21.5	68.6 ± 14.2	0.002
Median age (years, range)	70 (19–87)	55 (19–87)	71 (29–88)	-
Male gender (*n*,%)	46 (70)	12 (71)	34 (69)	0.926
ECOG-performance status (PS) (*n*,%)				
0	23 (35)	8 (47)	15 (31)	0.22
1	6 (9)	4 (24)	2 (4)	0.016
2	7 (11)	2 (12)	5 (10)	0.857
3–4	30 (45)	3 (18)	27 (55)	0.008
ECOG-PS (mean ± SD)	2.0 ± 1.7	1.1 ± 1.4	2.3 ± 1.7	0.015
ECOG-PS ≥ 2	37 (56)	5 (29)	32 (65)	0.01
Procedure of echocardiography (*n*,%)				
Transthoracic echocardiography	66 (100)	17 (100)	49 (100)	-
Transesophageal echocardiography	30 (45)	9 (53)	21 (43)	0.472
Detection of vegetation (*n*,%)	46 (70)	12 (71)	34 (69)	0.926
Infection site (*n*,%)				
Aortic valve	20 (30)	8 (47)	12 (24)	0.081
Mitral valve	35 (53)	9 (53)	26 (53)	0.993
Pulmonic valve	0	0	0	-
Tricuspid valve	7 (11)	0	7 (15)	0.099
Multiple valve involvements	6 (9)	2 (12)	4 (8)	0.656
Unknown	10 (15)	2 (12)	8 (16)	0.651
Naïve valve involvement (*n*,%)	53 (80)	17 (100)	36 (73)	0.018
Prosthetic valve involvement (*n*,%)	13 (20)	0	13 (27)	0.018
Prior dental work (*n*,%)	7 (11)	1 (6)	6 (12)	0.463
Past history of infectious endocarditis (*n*,%)	4 (6)	0	4 (8)	0.234
Underlying diseases (*n*,%)				
Heart disease	33 (50)	7 (41)	26 (53)	0.398
Chronic pulmonary disease	8 (12)	4 (24)	4 (8)	0.094
Diabetes mellitus	21 (32)	3 (18)	18 (37)	0.145
Chronic kidney disease	19 (29)	3 (18)	16 (33)	0.239
Hemodialysis	8 (12)	0	8 (16)	0.076
Hepatic disease	5 (8)	2 (12)	3 (6)	0.449
Collagen vascular disease	7 (11)	1 (6)	6 (12)	0.463
Cerebrovascular disease	12 (18)	1 (6)	11 (22)	0.147
Malignancy	16 (24)	1 (6)	15 (31)	0.04
Gastroesophageal reflux disease	3 (5)	2 (12)	1 (2)	0.097
Dementia	4 (6)	1 (6)	3 (6)	0.971
Charlson comorbidity index (mean ± SD)	2.1 ± 2.2	0.9 ± 1.3	2.5 ± 2.3	0.009
Charlson comorbidity index ≧ 3 (*n*,%)	19 (29)	2 (12)	17 (35)	0.072
Clinical symptoms (*n*,%)				
Immunological phenomena	23 (35)	5 (29)	18 (37)	0.585
Vascular phenomena	47 (71)	13 (76)	34 (69)	0.578
Heart failure	9 (14)	0	9 (18)	0.054
CNS disorder	17 (26)	4 (24)	13 (27)	0.807
Conditions of the patients (mean ± SD)				
SIRS score	1.8 ± 1.1	1.7 ± 1.1	1.9 ± 1.1	0.429
SOFA score	3.8 ± 2.7	1.9 ± 1.9	4.5 ± 2.7	<0.001
DIC	11 (17)	2 (12)	9 (18)	0.529
Shock state (SBP < 100 mmHg)	19 (29)	0	19 (38)	0.001
Persistent bacteremia *	12 (18)	2 (11)	10 (21)	0.456
Treatment (*n*,%)				
Surgical intervention	19 (29)	8 (47)	11 (22)	0.053
Initial antibiotic therapy (*n*,%)				
Monotherapy	25 (38)	6 (35)	19 (39)	0.799
Penicillins	10 (15)	2 (12)	8 (16)	0.651
Cephalosporins	7 (11)	2 (12)	5 (10)	0.857
Carbapenems	2 (3)	0	2 (4)	0.398
Anti-MRSA agents	2 (3)	0	2 (4)	0.398
Others	4 (6)	2 (12)	2 (4)	0.253
Combination therapy	41 (62)	11 (65)	30 (61)	0.799
Combination therapy with anti-MRSA agents	27 (41)	8 (47)	19 (39)	0.549
Combination therapy with aminoglycosides	10 (15)	3 (18)	7 (14)	0.739
Anti-pseudomonal agents use (*n*,%)	22 (33)	3 (18)	19 (39)	0.111
Duration of				
hospital stay (mean days ± SD)	64.2 ± 65.5	56.8 ± 32.7	66.8 ± 74.3	0.595
antibiotics use (mean days ± SD)	123.4 ± 179.6	131.8 ± 126.6	120.4 ± 197.8	0.827
Outcome				
Mortality (*n*,%)				
30-day mortality	8 (12)	0	8 (16)	0.076
In-hospital mortality	16 (24)	0	16 (33)	0.007
Inappropriate treatment (*n*,%)	9 (15) *	0	9 (18)	<0.001
Pathogens isolated (*n*,%) **				
*Streptococcus aureus*	23 (35)	5 (29)	18 (37)	-
MSSA	17 (26)	5 (29)	12 (24)	-
MRSA	6 (9)	0	6 (12)	-
Coagulase-negative streptococci	12 (18)	2 (12)	10 (20)	-
*Streptococcus anginous* group	5 (8)	2 (12)	3 (6)	-
Oral streptococci	10 (15)	3 (18)	7 (14)	-
*Enterococcus faecalis*	8 (12)	1 (6)	7 (14)	-
*Enterococcus faecium*	1 (1.5)	0	1 (2)	-
HACEK	3 (5)	2 (12)	1 (2)	-
Others	6 (9)	0	6 (12)	-
Multiple pathogens isolated (*n*,%)	10 (15)	0	10 (20)	-
Unknown (*n*,%)	4 (6)	2 (12)	2 (4)	-

DNAR, Do Not Attempt Resuscitation; HACEK, *Haemophilus* species, *Aggregatibacter* species, *Cardiobacterium hominis*, *Eikenella corrodens*, and *Kingella* species; ICU, intensive care unit; MRSA, methicillin-resistant *Staphylococcus aureus*; MSSA, methicillin-susceptible *Staphylococcus aureus*; PDR, potential drug resistant; RCT, randomized control trial; SD, standard deviation; SIRS, systemic inflammatory response syndrome; SOFA, sequential organ failure assessment. * Causative pathogens were unknown in 5 cases. Therefore, 16 and 45 cases were evaluated in RCT appropriate and inappropriate groups. ** A total of 72 pathogens were isolated from 66 patients.

**Table 2 jcm-12-01566-t002:** Reasons for rejection for enrollment into a randomized clinical trials.

Reasons	Number (%)
1. Age < 18 years	0
2. Coexisting comorbidities or medical conditions which could make the evaluation difficult	13 (20)
3. Prosthetic valve involvement	13 (20)
4. Immunosuppressive therapy	11 (17)
5. Chemotherapy	1 (2)
6. Palliative therapy	17 (26)
7. Hemodialysis	8 (12)
8. Having other complicated infection	7 (11)
9. Inability to give full informed consent	7 (11)
10. Causative pathogens are fungi	3 (5)
11. Multiple pathogens identified	10 (15)
12. Need of prolonged antibiotic therapy	2 (3)
13. Patients who need urgent cardiac surgery but are considered inoperable due to high surgical risk	9 (14)
14. Poor prognosis who are not expected to survive until the end of the trial	3 (5)
15. Pregnant or breastfeeding women	0

## Data Availability

All data generated or analyzed during this study are included in this published article.

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
