# Peer review of "A Gap of Patients with Infective Endocarditis between Clinical Trials and the Real World"

_jcm, 2023, doi:10.3390/jcm12041566_

Round 1

Reviewer 1 Report

I read the manuscript with interest. Here are some comments from this reviewer.

1. In several places, it is stated that the "RCT can never reflect the real-world population", but it should be modified to something milder such as "RCT reflects only a small proportion of the real-world population".

2. in the Abstract, it states that this study included patients between 2004 and 2019, but  Line 66 states between 2007 and 2019. Which is correct?

3. there are mixed expressions such as "RCT appropriate group" and "participation possible group", which need to be corrected.

4. has the patient's consent been obtained for this study?

5. please spell out what CCI is.

6. There are two blanks in Line 143.

7. please spell out  HACEK, etc in Table 1.

Translated with www.DeepL.com/Translator (free version)

Author Response

Response to reviewer 1

Manuscript ID: jcm-2186682

Type: article

Title: A gap of patients with infective endocarditis between clinical trials and the real world

Thank you very much for reviewing my article “A gap of patients with infective endocarditis between clinical trials and the real world.” I totally agree with your opinions and have revised the article according to your suggestions. The revised portions are in red in the text. I greatly appreciated your feedback and am positive that the revised article would be worth publication. Please, reconsider this article for publication in your journal.

Sincerely,

NOBUHIRO ASAI, and Hiroshige Mikamo on Feb 10, 2023

Reviewer 2 Report

The authors are conducting a retrospective study of 66 patients with infective endocarditis at their institution, focusing on patients in poor condition who would be excluded from a typical RCT. 49 patients were included, with older age, lower performance status, more PVE, more malignancies, higher SOFA scores, and more shock condition, and high SOFA score. And, as expected, they reported significantly worse survival. This is a valuable paper that clearly reports what is widely felt in actual clinical practice.

Major comment

(L70-72) The APACHE II score is commonly used to assess the severity of IE patients as sepsis (Circ J. 2019 Jul 25;83(8):1767-1809.) but I wonder why this is the method used in this study ?

(Table 1) Regarding surgical intervention, please mention how the indication for surgery was determined.

Minor comments:

(L138) The term "Native valve" should be used instead of "Naive valve" because it is commonly used in guidelines and other documents.

(L143) There are obvious mistakes in the description.

Author Response

Response to reviewer 2

Manuscript ID: jcm-2186682

Type: article

Title: A gap of patients with infective endocarditis between clinical trials and the real world

Thank you very much for reviewing my article “A gap of patients with infective endocarditis between clinical trials and the real world.” I totally agree with your opinions and have revised the article according to your suggestions. The revised portions are in red in the text. I greatly appreciated your feedback and am positive that the revised article would be worth publication. Please, reconsider this article for publication in your journal.

Sincerely,

NOBUHIRO ASAI, and Hiroshige Mikamo on Feb 10, 2023

Comments and Suggestions for Authors

The authors are conducting a retrospective study of 66 patients with infective endocarditis at their institution, focusing on patients in poor condition who would be excluded from a typical RCT. 49 patients were included, with older age, lower performance status, more PVE, more malignancies, higher SOFA scores, and more shock condition, and high SOFA score. And, as expected, they reported significantly worse survival. This is a valuable paper that clearly reports what is widely felt in actual clinical practice.

Major comment

(L70-72) The APACHE II score is commonly used to assess the severity of IE patients as sepsis (Circ J. 2019 Jul 25;83(8):1767-1809.) but I wonder why this is the method used in this study ?

A). We had previously reported that SOFA score was a good predictive marker among IE patents for in-hospital death. Thus, we used both SOFA and CCI. As you mentioned, the reason that SOFA score was used was not clear. I added the following sentence. Also, one more reference was placed in it. The APACHE 2 is overlapped with combined SOFA and CCI score. To avoid duplicate, we used SOFA score instead of APACHE 2 score.

In page 2,

Since we previously reported that SOFA score was a good predictive marker among IE patients for in-hospital death [7]. Thus, we evaluated severity of IE by SOFA score.  

(Table 1) Regarding surgical intervention, please mention how the indication for surgery was determined.

A). I do understand how you wonder. The indication for surgical intervention was not clear. As you suggested, there might have been a career-bias in physicians and surgeons. Thus, I added the following sentence in page xx. The sentence should be written as limitation.

In page 3,

There are several limitations in the study. First, this is a retrospective study with a small sample size at a single-center institute. Second, the study included only patients with definite IE according to the modified Duke criteria or by surgical procedure. Some IE patients might not have been included in our study. Third, there might have been a career-bias in physicians and surgeons. This might have affected the results.

Minor comments:

(L138) The term "Native valve" should be used instead of "Naive valve" because it is commonly used in guidelines and other documents.

A). Yes. It should be changed.

In page 3,

Results

A total of 66 patients were enrolled in the study. Table 1 shows the patients’ characteristics and clinical outcomes. They were 46 males (70%) and the median age was 70 years (range 19 to 87 years). As for the infection site, mitral valve was most commonly seen in 35 (53%), followed by aortic valve in 20 (30%). Native valve and prosthetic valve involvement were found in 53 (80%) and 13 (20%), respectively.

(L143) There are obvious mistakes in the description.

A). I am so sorry that I missed them. I added the correct numbers in the text as follows.

In page 3,

Results

A total of 66 patients were enrolled in the study. Table 1 shows the patients’ characteristics and clinical outcomes. They were 46 males (70%) and the median age was 70 years (range 19 to 87 years). As for the infection site, mitral valve was most commonly seen in 35 (53%), followed by aortic valve in 20 (30%). Naïve valve and prosthetic valve involvement were found in 53 (80%) and 13 (20%), respectively. In terms of underlying diseases, heart disease was most frequently seen in 33 (50%), followed by diabetes mellitus in 21 (32%). The mean CCI was 2.1 (±2.2). As for clinical symptoms, immunological and vascular phenomena were seen in 23 (35%) and 47 (71%), respectively. In terms of initial antibiotic therapy, monotherapy and combination therapy were seen in 25 (38%) and 41 (62%), respectively. The most common monotherapy was penicillin in 10 (15%), followed by cephalosporin in 7 (11%). Combination with anti-MRSA agents was used most frequently in 27 (41%). The mean duration of hospital stay and antibiotic treatment were 64.2 days and 123.4 days, respectively. The 30- and in-hospital deaths were seen in 8 (12%) and 16 (24%), respectively. Regarding pathogens, Staphylococcus aureus was identified most commonly in 23 (32%), followed by coagulase- negative streptococci in 12 (17%).

Round 2

Reviewer 1 Report

The questions raised by this reviewer were addressed accordingly by the authors.

Author Response

Response to reviewers for R2

Manuscript ID: jcm-2186682

Type: article

Title: A gap of patients with infective endocarditis between clinical trials and the real world

Thank you very much for reviewing my article “A gap of patients with infective endocarditis between clinical trials and the real world.” I totally agree with your opinions and have revised the article according to your suggestions. The revised portions are in red in the text. I greatly appreciated your feedback and am positive that the revised article would be worth publication. Please, reconsider this article for publication in your journal.

Sincerely,

NOBUHIRO ASAI, and Hiroshige Mikamo on Feb 12, 2023

Academic Editor Notes

Thanks for submitting this manuscript. It is well written. I have only some minor comments:
1) The first has to do with table 1. Numbers should be followed by a percentage in parenthesis (%) at all times (when applicable). For example, at the end of the table, in microbiology, percentages in parentheses are only shown in the first column.

A). I added the percentages in the table 1. The number s were corrected in the text.

2) Page 5 should be removed (there is a comma or dot in the middle)

A). Yes. I removed the space. I am sorry to bother you.

3) Author contributions could be added after the acknowledgments section, as per author guidelines.

A). I added the author contributions as follows.

In page 3,

Acknowledgments: We are grateful for the diligent and thorough critical reading of our manuscript by Dr. Yoshihiro Ohkuni, Chief Physician, Taiyo and Mr. John Wocher, Advisor, Kameda Medical Center (Japan). Also, we thank all medical staffs to help us care for these patients.

Author contributions: Conceptualization: NA, HM. Microbiological examination: DS and HS. Data collection: NA, YS, JH, HK, DS, HS and MH. Data analysis: NA, WO, HK, and MH. Original draft preparation: NA, Reviewing and editing: NA and HM. All authors read and agreed this article for publication.
